# Identifying Policy Actions Supporting Weather-Related Risk Management and Climate Change Adaptation in Finland

**Heikki Tuomenvirta [1,\*], Hilppa Gregow [1], Atte Harjanne [1], Sanna Luhtala [1], Antti Mäkelä [1], Karoliina Pilli-Sihvola [1], Sirkku Juhola [2], Mikael Hildén [3], Pirjo Peltonen-Sainio [4], Ilkka T. Miettinen [5] and Mikko Halonen [6]**

1   Finnish Meteorological Institute (FMI), P.O. Box 503, FI-00101 Helsinki, Finland
2   Ecosystems and Environment Research Program and Helsinki Institute of Sustainability Science (HELSUS), University of Helsinki, P.O. Box 65, FI-00014 Helsinki, Finland
3   Finnish Environment Institute (SYKE), Latokartanonkaari 11, FI-00790 Helsinki, Finland
4   Natural Resources Institute Finland (Luke), Latokartanonkaari 9, FI-00790 Helsinki, Finland
5   National Institute for Health and Welfare (THL), P.O. Box 95, FI-70701 Kuopio, Finland
6   Gaia Consulting Oy, FI-00120 Helsinki, Finland
\*   Correspondence: heikki.tuomenvirta@fmi.fi; Tel.: +358-50-574-6824

**Abstract:** Climate change adaptation (CCA) policies require scientific input to focus on relevant risks and opportunities, to promote effective and efficient measures and ensure implementation. This calls for policy relevant research to formulate salient policy recommendations. This article examines how CCA research may contribute to policy recommendations in the light of idealized set of knowledge production attributes for policy development in Finland. Using general background information on the evolution of CCA research and a case study, we specifically examine how the set of attributes have been manifested in research serving CCA and discuss how they have affected the resulting policy recommendations. We conclude that research serving CCA can be improved by more explicit reflection on the attributes that pay attention to the context of application, the methods of teamwork and a variety of participating organizations, transdisciplinarity of the research, reflexivity based on the values and labour ethos of scientists and novel forms of extended peer review. Such attributes can provide a necessary, although not sufficient, condition for knowledge production that strives to bridge the gap between research and policy.

**Keywords:** climate change; climate risks; weather risks; climate change adaptation; disaster risk management; national adaptation policy; policy implementation; policy recommendations; knowledge attributes

## 1. Introduction

Progress in the national planning of adaptation to changing weather risks and climate change has advanced rapidly in Europe. Many countries are currently pursuing different types of climate risk and vulnerability assessments after the publication of national adaptation plans and strategies [1]. The role of scientific research in the climate change adaptation (CCA) process has been recognized [2–5]. Hence, it is important to systematically improve our understanding of how CCA research is carried out and how it can serve policy development and decision making.

During the past 30 years, knowledge production supporting CCA policies has evolved from a focus on climate change as a physical phenomenon to examining the possible actions for CCA.

This stresses the need for a meaningful policy–research dialogue, which put specific requirements on the research.

CCA policy can be supported with dedicated research that is able to deliver syntheses that are salient from a policy making point of view [3,5]. Both academic and general applied research into climate change risks, impacts and vulnerabilities can provide a base for action [4]. However, research results may be either too general to translate into policies and measures or too specific to serve policy making that has to take into account a broad set of wider policy goals. Weather risks are numerous and different. The nature and magnitude of the risks that sectors are exposed to differ from one another. Hence, prioritization is needed to shortlist those risks calling for urgent action [6]. To support action, context specific knowledge from different sources is needed [7].

Commissioned research with clearly specified terms of reference is assumed to serve instrumental purposes but does not automatically deliver results that are feasible to implement [8]. Critical discussion on how to produce relevant knowledge has focused on the attributes of knowledge production. In particular, claims have been made that knowledge production should be increasingly based on reflexivity, transdisciplinarity and new modes of quality control [8]. The socio-political nature of CCA also calls for knowledge production that recognizes the context in which the knowledge is to be used [9]. However, empirical evidence that these attributes would have changed knowledge production or provided "better adaptation" is not particularly strong [10–13].

Our aims in this paper are to explore how research can improve CCA by reflecting on the process and results of a recent commissioned research project which we carried out to deliver knowledge for CCA in Finland. To place the project in a wider context we provide an overview of the development of knowledge production for CCA in Finland since 1990. We then use the project as a case study to discuss the fulfilment of idealized attributes of knowledge production for policy development. Our study provides insights into how multiple scientific and social dimensions as well as interactions and cooperation can be handled in practice.

## 2. Materials and Methods

We used the five knowledge production attributes explored by Hessels and van Lente [10] as a starting point to structure our analysis. The origin of the attributes can be traced back to the framework of "Mode 2 knowledge production" [14]. Hessels and van Lente [10] argued that the analysis of the constitutive attributes of policy relevant research should focus separately on the cognitive choice of the research agenda (*context of application*), the methods of teamwork and variety of organizations (*heterogeneity*), the epistemology socially robust knowledge including the organizational map of disciplines (*transdisciplinarity*), the values and labour ethos of scientists (*reflexivity*) and novel forms of extended peer review (*novel quality control*) (see Figure 1). For each of the attributes, we compiled data on Finnish CCA research and specific data from our case study, using a range of methods that are described in detail for each attribute in the respective sections.

The first knowledge production attribute, *context of application*, was examined by exploring the linkages between climate change research and policy making in Finland. The policy development and the characteristics of scientific information matching the policy process were explored [15]. The wider context where new knowledge for CCA need to be produced is described.

The other attributes were explored through a case study originally commissioned by the Government's analysis, assessment and research activities program managed by the Prime Minister's Office. The project 'ELASTINEN' concentrated on examining the state of weather and climate risk management and the role of different actors in Finland (The authors of this article were part of the ELASTINEN project team.). It analysed risk management measures and how actors related to these. In addition, it examined how the costs and benefits of risk management measures were assessed and how risk management can enable new business. The purpose of the project was to derive research-based recommendations and propose policy actions. The detailed results of the ELASTINEN project have been reported in Finnish-language reports [16–19].

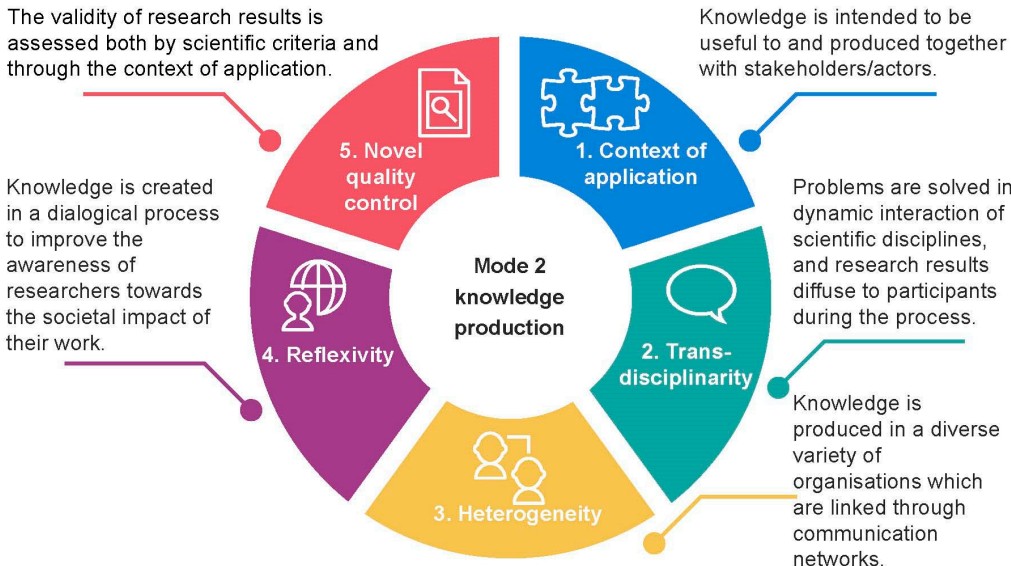

**Figure 1.** Mode 2 knowledge production attributes. Adapted from Hessels & van Lente [10] and Gibbons et al. [14].

## 3. Results

In the following, we analyse how the five knowledge production attributes have manifested in the Finnish research serving CCA. First, we examine the broad setting of the context of application and show that the ELASTINEN project is an outcome of this advanced, wide policy setting. Second, we use the project as a case for exploring how the other four knowledge production attributes of heterogeneity, transdisciplinarity, reflexivity and novel quality control have been manifested. The final section of this chapter presents the policy recommendations to enhance weather risk management and CCA in Finland.

### 3.1. The Context of Application: Policy-Science Interaction and the National Adaptation Plan

Hessels and van Lente [10] raised the context of application as a key attribute that stresses the links between knowledge generation and applications. Close links are expected to minimize the gap between science and policy. When examining this for Finland, it is necessary to also note the interaction with the European and wider international level. Using the cyclic framework of Vogel et al. [15] on science and policy-making interaction, relevant connections can be identified. However, we simplified the framework by showing only four stages of decision making (top row, Table 1), as adapted from IPCC work [5,20], to highlight the synchronic aspects of historical development.

The first national CCA-relevant research programs studied climate change, its causes, mechanisms and consequences and informed policymakers on CCA and climate change mitigation in Finland in the 1990s and early 2000s (Table 1). CCA appeared formally on the climate policy agenda in 2005 when the National Adaptation Strategy (NAS) was adopted as the first NAS in Europe [21]. During the 2010s, CCA policy increasingly moved toward the implementation stage, paralleling development elsewhere [22]. CCA is considered as a cross-cutting policy issue and national CCA policy is being developed jointly between ministries and implemented within the sectors, similar to many other EU countries [23,24].

Throughout the development of national CCA policy, there has been a continuous attempt to support the strategy formation and implementation with research programs and projects addressing sector-specific needs [2]. Several national research efforts have cooperated with their international counter partners, often through the European Union (EU) Framework Programmes' research projects. Synchronously, the supporting national and international research projects have focused, for example,

on finding and testing CCA solutions, finding synergies with mitigation and other policy objectives and defining new design standards and practices (Table 1).

The interaction between the international and national sphere takes place through coinciding policy and research agendas and interaction. Thus, the evolution of the Finnish adaptation policy reflects international climate policies from the UNFCCC (1995 onward), the EU adaptation strategy [25] and the Sendai framework on disaster risk reduction and the Paris Agreement, both from 2015. Similarly, the CCA research and development activities in Finland have developed in interaction with Nordic, European and international research. Several national research efforts have cooperated with their international counter partners, often through European Union Framework Programmes or research projects funded by the Nordic Council of Ministers. Some of the demonstration projects, for example, the European Union's Life Programme [26] and the Nordic Council of Ministers' Public Private Partnership for plant breeding [27] have also had international co-operation.

**Table 1.** Stages of decision making and the aims of supporting research and development aligned with climate change adaptation actions and examples of national research and evaluations in Finland as well as with international climate change adaptation (CCA) relevant research and development programs.

| Stage of Decision Making | Scoping Intelligence Gathering, Problem Definition | Policy Formulation Strategy Development and Promotion | Implementation Mobilization of Actors, Incentives, Routinization | Review and Development Monitoring, Evaluation, Learning |
|---|---|---|---|---|
| **Aims of research and development** | Development of scientific understanding and methodology, assessment of knowledge, awareness raising | Frame the problem, alter the goals, identify choices, collect new data, develop technology | Technology and other solution demonstration and launch, identification of obstacles, training | Development of indicators, performance data gathering and analyses, appraisal, promotion of good practices, mobilization of actors |
| **Policy action in Finland** | A National Adaptation Plan was requested by the Finnish Parliament in its reply to National Climate Strategy 2001 | National Strategy for Adaptation to Climate Change 2005 [21] | National Climate Change Adaptation Plan 2022 [28] Climate Change Act 2015 [29] | Mainstreaming of the policy objective: *"Finnish society has the capacity to manage the risks associated with climate change and adapt to changes in the climate"* |
| **Examples of research and evaluations in Finland** | SILMU Finnish research programme on climate change (1990–1995) | ISTO Climate change adaptation research programme (2006–2010) | REFI Climatological test years for building physics (2010–2013) | Evaluations of National Policies for Adaptation to Climate Change [24,30,31] |

The interaction between CCA policy making and the research community within Finland has been strong and has guided research. Research has evolved to increasingly deal with vulnerability and adaptation and researchers have been involved in the evaluation of adaptation policy. These evaluations have contributed to the development of the policy including the current National Adaptation Plan [24,30–32]. In addition, the administration has been active in outlining and funding research supporting policy-making. The role of scientific expertise in policy advice has further been strengthened and formalized by the Climate Act, according to which "the adaptation plan shall include a risk and vulnerability review, as well as action plans on adaptations specific to each administrative branch, if necessary" ([29], Section 8:2).

The most recent policy document outlining Finnish adaptation actions is the National Adaptation Plan 2022 [28]. Like the preceding national CCA policy document [21], the plan states that CCA is to be integrated into the planning and activities of relevant sectors and their actors. Actors are expected to have access to appropriate climate change assessments and risk management methods. This shall be achieved with research and development work and communication as well as education and training that can enhance the adaptive capacity of society, develop innovative solutions and improve the awareness of citizens on CCA [28]. One expression of this was the commissioning of the ELASTINEN project as part of the Government's analysis, assessment and research activities managed

by the Prime Minister's Office, a government initiative started in 2014 to "generate information that supports decision making, working practices and management by knowledge" [33]. The purpose of the project was to provide a synthesis of available research and to deliver policy recommendations. The project can therefore be used to examine how knowledge production attributes can be observed in practice.

*3.2. The Four Knowledge Production Attributes as Reflected in the ELASTINEN Project*

3.2.1. Transdisciplinarity and Heterogeneity

This section discusses how transdisciplinarity and heterogeneity were manifested in the ELASTINEN project. *Heterogeneity* refers to the diverse variety of organizations, universities, governmental research institutes and private expert and consulting groups that participate in research work [10]. Such heterogeneity was built in the ELASTINEN consortium, which included partners from several government research institutes, a university and a consultancy firm.

Transdisciplinarity refers to the mobilization of a range of theoretical perspectives and practical methodologies to solve problems [10], which in this project was to derive policy recommendations. This was reflected in the ELASTINEN project in the analysis of meteorological and hydrological extreme events and their impacts, the review of projections of future climatic extremes, surveying weather risk management, examining the use of economic analysis methods to choose CCA measures, studying cross-border effects of climate change and examining opportunities for developing know-how in risk management and other business activities [16]. The relevant CCA research questions were co-created by the administration and the researchers. The analytical work, especially the development of policy recommendations, required more than a multidisciplinary combination of findings from different sectors. The transdisciplinarity analyses focused on common patterns, searched measures addressing the barriers and gaps identified and searched for findings that could develop the awareness of CCA activities.

To understand and evaluate how weather and climate risks are managed in different organizations, the risk perceptions and risk management practices including the use of climate information were explored using surveys and a stakeholder workshop [17]. Two parallel online surveys were conducted to study how weather and climate risks were perceived and managed, one for Finnish organizations in general and the other specifically for municipalities. The municipality survey (detailed in Reference [7]) was sent to 122 municipalities that have been active in climate change planning and received 33 responses. The questionnaire for public and private organizations was sent to 500 recipients, of which 118 responded (detailed in Reference [34]). The surveys were conducted using online forms that were open for two weeks (30 November–13 December 2015). The data from both surveys are stored in the Finnish Social Science Data Archive [35].

The surveys were exploratory and the respondents did not represent an unbiased sample of Finnish organizations or municipalities. Therefore, mainly qualitative conclusions can be drawn. There was, for example, a dominance of responses from the agriculture and food production and health sectors. The responses do, however, cover a wide range of activities (Figure 2). A workshop was also organized to gain further insights into the responses. Participants were invited from the same sample that was used in the surveys. Forty-two stakeholders and experts participated in a one-day event (27 January 2016). The survey results were jointly interpreted and ideas for the future development of risk management and information services were collected. This provided a transdisciplinary base for observations that supported the development of policy recommendations. For example, the insight that the main drivers for weather and climate risk management vary significantly between private and public organizations highlights the need for differential communication of CCA. The private sector emphasizes the avoidance of economic losses as the principal driver of weather and climate risk management, whereas for municipalities, securing vital functions and maintenance of infrastructure are the main drivers.

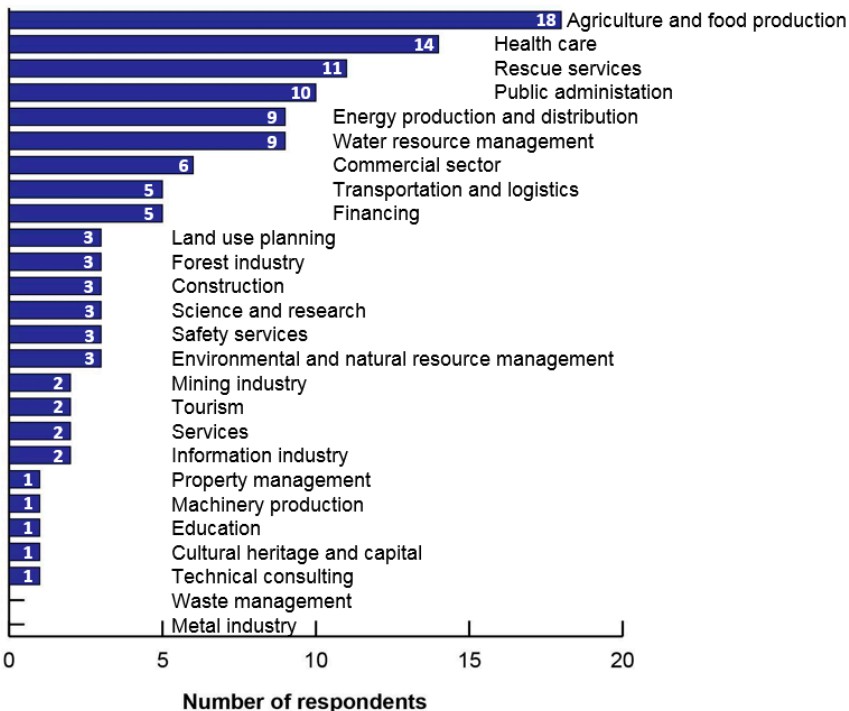

**Figure 2.** Organization survey respondents by sector. Roughly half (53%) were public organizations, 38% represented private enterprises and the remaining 9% were non-profit organizations.

The different, multidisciplinary analyses and assessments of the project provided a basis for transdisciplinarity. An analysis of the use and usability of economic analysis methods for the cost efficiency appraisal of various risk management and adaptation measures in Finland was performed [18]. Based on a literature review, six decision support methods were examined and their use in Finland analysed. Document analysis and expert interviews were used to examine the methods used in flood risk management, water services and urban planning. A social cost benefit analysis on the weather-related requirements of the revised Electricity Market Act (2013) was also conducted [36]. The analysis identified, for example, a lack of "administrative incentives" for allocating resources for economic analyses on risk management and CCA measures, although this should be done particularly when the implementation of measures leads to major investments.

An example of transdisciplinarity work was the exploration of the knowledge gap concerning the cross-border (transboundary) effects of climate change in Finland. Cross-border impacts are examined over multiple sectors by exploring impact chains related to events that are initiated elsewhere in the world but which also create impacts in the target country [37]. ELASTINEN examined the available literature on the global impacts of climate change, the statistics on the international connections of the Finnish society and undertook interviews with relevant actors about the significance of the cross-border impacts on different sectors [19]. The analysis concluded that the current deficiencies in the risk assessment and management could be addressed through cooperation and regular dialogue between sectors and countries.

Another area of transdisciplinary work was the exploration of commercial opportunities related to climate change. The survey (see above) revealed that many companies have recognized the need to take action in the face of new weather and climate risks. This includes improving the management of weather-related risks and acknowledging that the changing frequency and severity of harmful weather events require improved risk mitigation via new technology and procedures or increased risk sharing. Some of these actions may provide competitive advantages through cost savings, strengthened business continuity and reputational benefits. Some private sector actors plan to proactively seek competitiveness benefits or seize business opportunities in Disaster Risk Management and CCA [34].

Finland has been active in the capacity building of weather forecasting and warning services and CCA in developing countries [38,39]. Public, private as well as third sector organizations have gained experience in implementing these development projects. Our analyses showed that there is business potential in weather and climate related risk management and CCA in Finland as well as internationally, which can be supported with national adaptation policy [16]. There are other areas such as agriculture that may benefit from climate change but in which so far only modest CCA actions has been taken in Finland (earlier sowing, novel crops depending on region and later maturing cultivars [6,40]). Additional dedicated investments have not been made [41,42]. Thus, there may be some untapped opportunities.

### 3.2.2. Reflexivity and Novel Quality Control

This section highlights how reflexivity and novel (type of) quality control of knowledge production were reached. Reflexivity refers to the dialogical process of knowledge creation and how this also increases awareness of research results and their application. Reflexivity requires sufficiently wide and active participation and heterogeneity. It is a key component of policy-relevant knowledge production.

To ensure reflexivity in ELASTINEN, stakeholders from public, private and third sectors were encouraged to participate in generating information, contributing to a strong connection between CCA and other policies and (subjective) risk perception. Reflexivity was thus supported by engaging with two major voluntary-based organizations, the Finnish National Rescue Association, which coordinates the voluntary fire brigades in Finland and the Finnish Red Cross, which has over 30,000 volunteers across the country and coordinates the Voluntary Rescue Service. The reflexive process identified major barriers, knowledge gaps and good practices of weather-related risk management and CCA, leading to the identification of actors and the discovery of underused skills and sources of information [16].

The novel quality control allows for an assessment of the validity of scientific results that is directly related to their application. Hemlin and Rasmussen [43] also interpreted this to include ongoing evaluation processes during the research work. In the project, the quality control was novel in that it included reviews by CCA practitioners.

In the case of the ELASTINEN project, both reflexivity and novel quality control were seen in the interactive synthesizing of research findings and deriving policy recommendations. This included collaboration with many stakeholders and influenced the way in which policy recommendations were derived from data and the trans- and multidisciplinary findings (Section 3.2.1).

The reflexivity supported a deeper recognition of the role of the finance and insurance sectors to provide resources and facilitate weather and climate risk management. Large financial service actors have funded customer-side CCA measures that improve the resilience to weather hazards and, thus, reduce the risk of losses [44]. A key finding was that the Finnish CCA policy would benefit from more active participation of the financial service sector [16].

### 3.3. Policy Recommendations to Improve the Management of Weather Risks and Advance Climate Change Adaptation

A crucial methodological component for the successful formulation of policy recommendations was the wide participation and input from the stakeholders. These included policy and decision makers at different spatial levels of administration, experts, trade associations, companies and third sector organizations. In the beginning, the project involved a wide spectrum of stakeholders via online surveys and interviews and through a workshop. Similarly, the final seminar was organized for a wide audience of stakeholders. Involvement was supported by blog texts and news during the project that were communicated through the project website and social media (Facebook and Twitter).

To gain insightful input, the advisory group of the ELASTINEN, the "Monitoring Group on the Adaptation of Climate Change of Finland," was invited to comment and approve the policy recommendations and the final report of the project [16]. The group consisted of representatives from the relevant ministries and other authorities, regional and local administration and research institutes.

Draft versions of the policy recommendations were also presented at the final stakeholder seminar of the project. At the seminar with 99 attendees, an expert panel (participants from the finance, energy and third sectors, an innovation fund, parliament and government ministry) had an enlightening discussion on the management of weather and climate risks. The comments from the advisory group and stakeholders helped to focus the concise recommendations and proposed policy actions.

The project's key findings and policy recommendations were synthesized in a concise, 32-page report to maximize the potential readership of the report. Furthermore, a policy brief was prepared [45]. The policy brief specifically targeted municipal policy and decision makers because the ministry responsible for CCA in Finland considered local level risk management important and of requiring increasing effort.

Three groups of policy recommendations were provided: (1) the production and use of information should be diversified, (2) cooperation between different sectors and actors should be strengthened and procedures improved and (3) services and business opportunities should be developed. To make the general recommendations actionable, altogether eight main and 22 detailed policy actions were proposed. In addition, their potential benefits were identified and the actors responsible were identified. Actors included research institutions, service providers (e.g., weather and risk information), universities, research funders, private sector actors (inter alia finance and energy sectors), trade associations, networks that promote and support business, the Government, public administration at different levels, public sectors (such as rescue services, healthcare) and non-governmental organizations. Detailed recommendations and the main measures proposed by ELASTINEN are presented in Appendix A.

## 4. Discussion

Our findings showed that CCA research and policy making in Finland have co-evolved. This has led to demand for research-based policy recommendations to enhance the implementation of the current CCA plan in Finland. The research community has been able to respond to the demand because it has an understanding of the policy processes. This interaction fits the idea of "Mode 2 knowledge production" including the five attributes of context of application, transdisciplinarity, heterogeneity, reflexivity and novel quality control [10,14] (Figure 3).

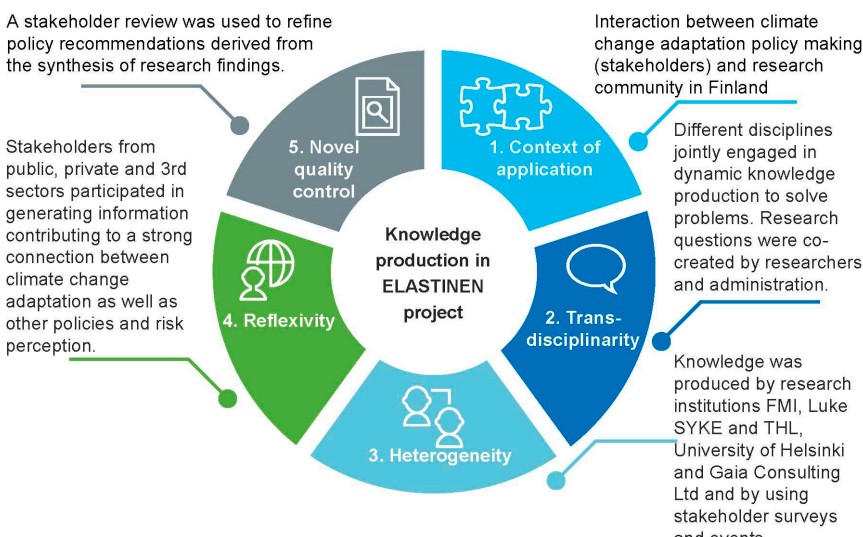

**Figure 3.** Characteristic examples of the knowledge production attributes applied in the ELASTINEN project. Schematic figure is adapted from Hessels & van Lente [10] and Gibbons et al. [14] (see Figure 1).

We argue that knowledge production and the provisioning of policy advice have developed in such a way that the five attributes can be identified in the broad overall context (Section 3.1) and in the specific case (Sections 3.2 and 3.3). As we were part of the team that implemented the project, it has

been easy to trace the attributes. However, it also means that there is a risk of over interpretation in the search for attributes. We have made an effort to avoid this by using verifiable observations as much as possible in identifying the attributes in Section 3.

There is no universal agreement on the relative relevance of the five attributes but they have appeared with different weights in various efforts aimed at explaining the emergence of societally relevant research [10]. There are no agreed criteria for when knowledge production attributes have been sufficiently fulfilled but the following successes and deficiencies can be identified for the Finnish case.

The context of application has emerged gradually as a result of more than twenty years of cooperation and co-development between the research community and decision makers in setting agendas for research programs. Finland has most likely benefited from being a small country with low hierarchies and many opportunities for personal contacts between researchers and policy makers. Funding has been available and used to guide research to tackle policy relevant topics, also internationally. Over the years, policy needs have thus influenced the aims of the research. Conversely, research results have influenced CCA policy framing. The sectorial approach of administration can, however, still be cumbersome and inefficient, especially with regard to cross-cutting issues, as shown by a recent study [23]. In addition, earlier findings that there are remarkable differences between sectors in the speed of CCA policy progress still seem to be valid [24,30].

The heterogeneity of the ELASTINEN project team (research institutes FMI, SYKE, Luke and THL; University of Helsinki and Gaia Consulting Ltd., see author affiliations) enabled contacts with a wider set of sectors and actors than before (including the private sector and the third sector). The stakeholders provided essential input, for example, they could confirm the survey result that the majority of the Finnish organizations, especially in the private sector, focus on near-term (<5 years) weather risks rather than long-term climate risks. This finding guided the project to use a wider framing of the research to include both weather and climate change risks. It highlights the opportunities for improved linking of the national agenda setting with local implementation through the integration of weather risk management and CCA to achieve potentially wider implementation and uptake of strategies and plans [46].

The research questions of ELASTINEN were prepared in a transdisciplinarity manner in a Government working group for the coordination of research, foresight and assessment activities. It used input from expert groups (a mix of administration and research experts) as well as ideas from individual researchers. In addition, the transdisciplinarity working method of the research team allowed the combining of the results of different types of information gathering approaches (surveys, interviews, literature review and document analysis of several disciplines, as well as statistical, interpretation of climate modelling results and other quantitative analyses). A major obstacle for the transdisciplinarity work was the limited project funding (350 k€), as this limited both possibilities of using researcher and stakeholder expertise.

Preliminary results were elaborated with stakeholders in a reflexive manner. Stakeholders stressed the applicability and accessibility of results to be developed. It became clear that the timing of information services needed to match the decision processes of the end-users. This may be problematic both in the private and public sector because, although CCA research is accumulating continuously, it is collaborative research projects that provide opportunities for comprehensive, reflexive knowledge production.

The main criteria of novel quality control outside the traditional scientific approach was the feedback from the "Monitoring Group on the Adaptation of Climate Change of Finland" and participants of the final seminar. To create stronger awareness of the research results and their application, both reflexivity in the derivation of recommendations and the role of actors in quality control could have been more extensive during the project. This finding is included in the recommendation 'to intensify and enlarge networks relevant to CCA and weather risk management.'

It is difficult and for long-term changes too early, to estimate the role of ELASTINEN's recommendations in achieving the policy objectives of CCA in Finland. Impacts on policy by a

particular research initiative are necessarily subtle. Some recommendations have, however, already been implemented. Thus, the "Monitoring Group on the Adaptation of Climate Change in Finland" has been enlarged to include representatives from the financial and third sectors. The financial sector in Finland has also developed a joint climate policy and outlined an approach for improved climate risk disclosure. A key driver has probably been the active participation in the work of the Financial Stability Board's Task Force on Climate-related Financial Disclosures [47]. The ELASTINEN recommendations to the finance sector may, however, have contributed to raising awareness in Finland. A national vulnerability and risk assessment has also been carried out [48], leading to a proposal for a common governance model for climate risk assessments [49]. These experiences support the observations of Rau et al. [50] that intangible research impacts concerning possible shifts in opinion and practices among key policy and civil society actors should be recognized in determining the merit of societally oriented research.

Except for the attributes that we have explored, the attributes of credibility, relevance and legitimacy of knowledge are widely recognized and often referred to in the characterization of effective science–policy interfaces (e.g., [51,52]). These are related to the attributes that we have used. Thus, credibility is linked with quality control but also reflexivity; relevance is related to the context of application and transdisciplinarity; and legitimacy with reflexivity. Furthermore, Dunn and Laing [53] suggest that many attributes reflect the ethos of researchers, whereas policy-makers value applicability, comprehensiveness, timing and accessibility. These attributes reflect a need for operational and instrumental knowledge. Applicability and accessibility were also raised by the stakeholders of ELASTINEN.

Heink et al. [54] and Hansson and Polk [55] have pointed out that the attributes are generally difficult to evaluate and that evaluations are therefore rare. Despite the challenges related to the evaluation of attributes, we argue that analyses of their practical manifestation should be encouraged. The attributes focus on aspects of the processes of knowledge production that should be strengthened in the science-policy dialogue. Through a reflexive analysis of their fulfilment, as has been done in this study, it is possible to identify areas requiring specific attention and to improve future research supporting CCA. The attributes can provide a necessary, although not sufficient, condition for knowledge production that strives to bridge the gaps between research and policy. An emphasis on such attributes is particularly justified in research programs such as the national and EU Framework Programmes that aim to support CCA policies.

**Author Contributions:** HT, HG, SJ and MH conceptualized this manuscript. HT and SJ focused on the methodology. HT, SJ, MH, SL and HG prepared the original draft. HT, AH, SL, KPS, SJ, MH, PPS, ITM and M. Halonen reviewed and edited the manuscript. AH and AM were responsible for data curation. All authors investigated. HG and SJ were responsible for the funding acquisition and HG on project management and supervision. All authors read and approved the final manuscript.

**Funding:** This research was funded by the Government of Finland's analysis, assessment and research activities for 2015. AH was supported by the Jenny and Antti Wihuri Foundation.

**Acknowledgments:** The authors would like to thank our colleagues on the ELASTINEN project team for their work: T. Carter, F. Groundstroem, R. Haavisto, S. Haanpää, J. Jakkila, A. Jurgilevich, A. Kokko, V. Kollanus, T. Lanki, V. Nurmi, K. Oljemark, A. Parjanne, A. Perrels, A-J Punkka, T. Raivio, A. Räsänen, K. Säntti, N. Veijalainen and O. Zacheus. We would also like to thank Chairperson S. Lilja-Rothsten and the whole "Monitoring Group on the Adaptation of Climate Change in Finland" for their guidance and cooperation during the ELASTINEN project.

**Conflicts of Interest:** The authors were part of the ELASTINEN project team. The ELASTINEN project was originally commissioned and funded by the Government's analysis, assessment and research activities program managed by the Prime Minister's Office. Funding organization had no role in the collection, analyses or interpretation of data, in the writing of the manuscript or in the decision to publish the results.

## Appendix A

Policy recommendations and actions to improve weather and climate related risk management and to support adaptation efforts in Finland were derived (see Table A1).

**Table A1.** Summary of policy recommendations including proposed policy actions to improve the management of weather and climate risks and climate change adaptation, the potential benefits of the recommendations and the key actors (adapted from Gregow et al. 2016).

| Recommendations | Proposed Policy Actions | Benefits | Key Actors |
|---|---|---|---|
| **Diversify the production and use of information** | Gather and share information about weather and climate related risks more widely by<br>- allocating resources to impact studies<br>- creating incentives for open data<br>- developing a weather impact database<br>- assessing climate risks and vulnerabilities, especially health | Enhanced collection, sharing and implementation of information and knowledge | Research institutions, universities, private sector actors (especially financial services and energy), public administration of different sectors and at different levels, non-governmental organizations |
| | Monitor cross-border impacts of climate change regularly that is, every 4–5 years by<br>- putting an emphasis on key sectors with strong links to global markets or neighbouring areas<br>- assessing implemented measures | Develop foresight and plan policy to mitigate impacts | Government and its analysis, assessment and research activities, research institutes and universities |
| **Develop services and business opportunities** | Develop new tailored weather and climate services, e.g.,<br>- impact-based weather forecasts and warnings<br>- 10–30 days climate impact outlooks<br>- interactive tools/databases to support adaptation | Fittingly resourced and well-timed measures | Research and service organizations, private actors, funding agencies for research and development, trade associations, rescue services |
| | Support business opportunities related to adaptation by<br>- mapping the potential and strengths of Finnish know-how in international "adaptation market"<br>- integrating risk management know-how into research and development initiatives, management products and services | More diverse export and employment | Networks to promote business opportunities and trade, funding agencies for research, development and innovation |
| **Strengthen cooperation and improve procedures** | Develop and maintain cross-sectoral network to improve the weather risk management by<br>- allocating resources to the network facilitators<br>- keeping the network open for new actors<br>- organizing events to identify national and international business opportunities related to adaptation | Improved management of weather and climate related risk, seizing opportunities | Government, rescue services, health care, actors facing weather and climate risks, actors in charge of national security, research and service organizations, NGOs, business promotion networks |
| **Strengthen cooperation and improve procedures** | Promote economic assessments in decision making by<br>- developing guidance and instructions<br>- developing tools for assessing the impacts of risks and the costs and benefits of risk management measures<br>- conducting detailed economic assessment of risk management measures especially for large-scale investments | Better assessment of adaptation options and more cost-efficient solutions | Actors in the public and private sector responsible for the planning and implementation of weather and climate risk management and CCA |
| | Strengthen the awareness of financial services on their role in improving the management of weather and climate risks by<br>- activating existing networks on weather and climate risks<br>- mapping the measures of international pioneer countries towards financial services on the subject | Wider use of knowledge potential and resources in risk management | Actors in financial services, such as banking, asset management and insurance sector and authorities supervising financial services |
| | Participate in international adaptation efforts by<br>- giving resources into international processes related to weather and climate risks<br>- assessing national measures to promote the Sendai process<br>- taking cross-border impacts of climate change into account in development cooperation | Step forward as an international adaptation operator | Government ministries responsible for foreign, interior and environmental affairs |

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
