# Peer review of "Identifying Policy Actions Supporting Weather-Related Risk Management and Climate Change Adaptation in Finland"

_sustainability, doi:10.3390/su11133661_

Round 1

Reviewer 1 Report

I can provide minor comments for this paper. 

Line 46-48 - Provide a reference. 

Line 54-55 - Provide a reference. 

Figure 2. You can put also the number for each respondent by sector, even if are on the orizontal axis. 

Author Response

We thank reviewer for comments.  References have been added to the sentences presenting statements (lines 46-48 and 54-55 in the first version of manuscript). We have also redrafted Figure 2 based on wishes of the Reviewer.

Reviewer 2 Report

The current paper presents a very good example on how climate change adaptation policy could be shaped through actions as information gathering, analysis and stakeholder involvement, using as a case-study, actions and policy development in Finland. The paper is of very high interest for the audience Sustainability reaches out to, and very well written. My sole recommendation would be to expand the introduction section with additional, specific information on the course of the evolving of knowledge on how to address Climate Change Adaptation over the past decades, which the authors only briefly reference.

Author Response

We thank reviewer for comments. We have redrafted Abstract. In a more balanced way the Abstract summarizes the main conclusions. Also the scope of paper is clarified. We have also corrected language.

Reviewer 3 Report

This study is to identify policy actions regarding support weather-related risk management and climate change adaptation in Finland. One of the conclusions is the communication with policy makers and other stakeholders can improve cost efficient disaster risk management. I have some comments listed below to improve this manuscript.

1, In the abstract, it is would be good to address what is the ideas that would be good for testing in other countries and regions that are going through similar phases of climate change adaptation policy planning and implementing.

2, Is this a review paper or a research paper? The manuscript did an overview of the development of knowledge production in Finland since 1990, and examined a recent research project. It is not clear to me whether this manuscript could be characterized as a review paper or not? Two figures in the article are from other studies. It is needed to make this clear.

3, The limitations of this study, the implications of the main conclusion should be addressed more in detail and in depth. Please elaborate more regarding any better way or suggestions for better communications?

4, Line 305, ‘the project’s key findings and policy recommendations were synthesized in a concise, 32-page report to maximize the potential readership of the report.’ The readers would be interested in the key findings here.

Author Response

We thank reviewer for comments.

1)      We have redrafted Abstract so that it in a more balanced way summarize the main conclusions. Conclusions in the Abstract are now general, i.e. not country specific.

2)      We have modified text to explain why there is broad (“review style”) overview of knowledge production in Finland with linkages to international research and policy. This is to describe the context where case study, ELASTINEN project, was commissioned and performed. Analysis of the first of the five knowledge production attributes “context of application” requires a thorough examination of the wider research-policy interactions that form also a timeline of progress in CCA. We justify the selection of figures and tables:

- We present the first ever schematic diagram of very widely used framework of Gibbons et al. (see section Materials and methods).  To improve the readability of the article we moved the application of the framework in the beginning of section Discussion as it nicely summarizes main characteristics of this study and at the same presents them in the framework. The color of fonts have been changed to make figures 1 and 3 to look visually different form each other.

- Figure 2 was modified based on Reviewer 2 comment. The data has not been presented before in peer-reviewed, English language publication.

- Table 2 and A1 present analyses performed for this paper.

3)      The rewritten Abstract and modifications of the body of the article now more clearly present that the focus of the paper is examination of knowledge production for the CCA, especially, study of processes with attributes widely used. Similarly the main finding and limitations are presented based on examination of attributes. The ELASTINEN project results contain more detailed and practical findings and suggestions for advancing CCA in Finland in the present (societal, political, research, etc.) context. Unfortunately we feel that the scope of this paper is not fitting to expand into details and recommendations more than to include the Appendix.

4)      Reader is directed to the section describing shortly the key findings. The focus of the article is knowledge production but we assumed that some of the readers might be interested in the actual policy recommendations produced. For policy recommendations the reader is guided to study the Appendix.

Round 2

Reviewer 3 Report

This version is good.